# An Overview of In Vitro Drug Release Methods for Drug-Eluting Stents

**DOI:** 10.3390/polym14132751

**Published:** 2022-07-05

**Authors:** Navideh Abbasnezhad, Nader Zirak, Stéphane Champmartin, Mohammadali Shirinbayan, Farid Bakir

**Affiliations:** 1Arts et Métiers Institute of Technology, CNAM, LIFSE, HESAM University, F-75013 Paris, France; nader.zirak@ensam.eu (N.Z.); stephane.champmartin@ensam.eu (S.C.); 2Arts et Métiers Institute of Technology, CNAM, PIMM, HESAM University, F-75013 Paris, France; mohammadali.shirinbayan@ensam.eu

**Keywords:** in vitro release testing, drug-eluting stents, flow conditions, hydrogels, IVIV correlation

## Abstract

The drug release profile of drug-eluting stents (DESs) is affected by a number of factors, including the formulation, design, and physicochemical properties of the utilized material. DES has been around for twenty years and despite its widespread clinical use, and efficacy in lowering the rate of target lesion restenosis, it still requires additional development to reduce side effects and provide long-term clinical stability. Unfortunately, for analyzing these implants, there is still no globally accepted in vitro test method. This is owing to the stent’s complexity as well as the dynamic arterial compartments of the blood and vascular wall. The former is the source of numerous biological, chemical, and physical mechanisms that are more commonly observed in tissue, lumen, and DES. As a result, universalizing bio-relevant apparatus, suitable for liberation testing of such complex implants is difficult. This article aims to provide a comprehensive review of the methods used for in vitro release testing of DESs. Aspects related to the correlation of the release profiles in the cases of in vitro and in vivo are also addressed.

## 1. Introduction

### 1.1. Evolution of Cardiovascular Stents

There has been a global increase in the number of patients being treated each year for Cardio Vascular Disease (CVD). It is increasing, due to risk factors such as smoking, diabetes, obesity and lifestyle changes. According to the World Health Organization (WHO), in 2016, 17.9 million people died from CVDs, accounting for 31% of all deaths worldwide. Following this issue, the first developed technology, balloon angioplasty, was introduced. The very first angioplasties, based on balloon expansion in the artery, faced the problems of elastic recoil and neointimal hyperplasia and were the reason for 40–60% of restenosis in the years 1977–1990 [1,2]. Figure 1 shows the phenomenon of restenosis after angioplasty with only a balloon.

It is noteworthy that there are second generations of balloons coated with drugs, commonly paclitaxel, to overcome in-stent restenosis. In this case, the drug should be transferred rapidly during the contact of the balloon with the vessel wall, which lasts approximately one minute. Some researchers are interested in this technique as this method decreases the risk of bleeding, avoids the risky presence of a foreign object in the body and limits the side effects [4]. However, this method was not completely developed for various reasons, mainly due to the promising results of stents.

Furthermore, due to the restenosis of the treated artery after deflecting the balloon, Bare Metal Stents (BMS) were introduced in the 1990s to overcome to this deficiency in the ballooning, and line the artery wall, wherein the incidence of restenosis decreased to 20–30% due to the elimination of elastic recoil between 1991 and 2003. Meanwhile, as a disadvantage, BMS acted as a foreign object for the immune system and this may respond to this intrusive object in a variety of ways: macrophages (white blood cells) accumulate around the stent, and nearby Smooth Muscle Cells (SMC) proliferate, disrupting the process of endothelialization; migration and proliferation of vascular SMC from the media to the intima, generating an extra cellular matrix layer in the intima (intimal hyperplasia) followed by a narrowing of the luminal area (see Figure 2) [5,6]. It is likely that the phenomenon of thickened intima is due to the leukocytes that adhere to the activated endothelium and disrupt its recovery [7,8,9,10,11,12,13].

Therefore, the incorporation of the stents in the body was achieved by combining the drugs to avoid these serious effects. Thereafter, the treatment of the restenosis in the artery has been revolutionized by new generation of the stents, named drug eluting stents (DES) in 2003 [14]. This generation carries restorative drugs with them in order to treat the stenting area. These stents were coated by a polymer layer containing an active substance used to reduce neointimal hyperplasia. The incidence of restenosis decreased to about 3–20%. Although the use of DES rather solved the problem of restenosis, the issues of denuded intima and the related inflammation and thrombosis (shown in Figure 3) still persist and open a wide range of research [8,15,16,17].

It is reported in 2010 that annually about 0.3–0.6% of stenting with DES is followed by stent thrombosis (ST) followed by increased human mortality by 10–30% [18]. Stent malapposition, late or incomplete re-endothelialization and polymer induced inflammation are the main reasons for the inflammation and late thrombosis [5,8,16,19,20,21,22].

To overcome this problem, scientific advances are being made. They started about 40 years ago and the techniques continue to improve today: balloon angioplasty, bare stent, drug-eluting stent, bioresorbable stent. Figure 4 summarizes the different stents from the material view [19,23,24,25,26,27,28,29,30].

The feasible way to decrease the ST is to prevent the risk of bleeding after stenting using anticoagulants and antiplatelet agents (the healthy endothelium also affords the anti-inflammatory support due to natural anticoagulant protein C [8]. Maintaining the dose of drug during the therapy can minimize the risk of thrombosis [31].

### 1.2. Laboratory Methods for the Development of DES

Drug distribution in the arterial wall depends on many parameters such as the type of drug and its initial concentration, drug release rate into the arterial wall, drug solubility, particle size, binders, wetting, properties of the polymer matrix, coating methods, eluting direction, coating thickness, pore sizes in the coating, release conditions (release medium, temperature, pH), Reynolds’s number, blood flow kinetics, etc., [6,7,32,33]. Optimizing these parameters and investigating their effects can improve the kinetics of the drug release during the therapy.

In this aim there are three types of test environments. The most difficult and costly tests are related to the in vivo tests, which are performed on the living organism of the human body or animal body. The advantages are in having the real conditions of the test environment (real tissue and real medium), however, the covenant is that we sacrifice the life of the test animals, and are complicit in having access to the site of the therapy for local analysis and characterize the drug carrier during the release time. Moreover, the repeatability and testing assumptions confront some limitations. In vivo experiments usually focus on the levels of drugs in the blood instead of the concentration of drugs in the wall of the blood vessel, although the former is the aim of the therapy [34].

However, the drug concentration in the tissue can only be calculated following stent removal in animal models. Actually, the in vivo tests are only favored when the certitude of the drug carriers has already been totally proved, by non-living organisms.

Therefore prior to in vivo test, ex vivo experiments are categorized for biological research. In this condition, the medium stays as an artificial biological medium, but has the advantages of the real conditions of the tissue layer, such as the real curvature and dimension of the artery, the coefficient diffusion of the tissue, binding–unbinding of the drug with the receptors. 

In addition, using the ex vivo environment has been one of the most effective ways to analyze the physical, chemical, mechanical, and other properties of the tissue in interaction with the implanted biomaterial. Although these methods cannot exactly replicate the conditions that occur inside a living organism, they can mimic the in vivo conditions and lead to reliable testing results by providing controlled environments. For these kinds of tests necessary attention should be considered in order not to rot the donor organism.

The other common method is the in vitro experimentation. This method is prior to the other methods. The easiness of the test and manipulation, low cost, and accessibility of the facilities of the experimentation compared to the other methods have encouraged most laboratories to work with it. It is well known that increasing the accuracy of in vitro tests can reduce the sacrifice of a living body as well as the cost of in vivo testing. Therefore, reproducing real environmental conditions remains an enormous challenge for the researchers. 

Providing a test facility for the investigation of the drug delivery systems has been one of the most important steps in analyzing the different carriers. The availability of the equipment, the repeatability of the test as well as the final price of the used system, are among the important parameters in the construction and consideration of a system for studying the drug release systems. 

Therefore, an in vitro test apparatus that can provide an accurate estimate of the in vivo condition will be of great importance. The challenge in the in vitro method is to approach the test conditions to the more realistic conditions. Numerous test facilities are developed so far for the studying of the drug release. Among the different parameters of the test conditions, media which will represent the blood, porous media representing the tissue layer, type of the flow and its circulation in the bio-relevant system, which would be in the place of the cardiovascular system, should be optimized in order to best represent the characteristics of the in vivo environment.

This article will provide a comprehensive overview of the different aspects of the in vitro method for developing drug-eluting stents.

## 2. Different Geometrical Models of Drug-Eluting Stents

The studies show that in most of the cases, for studying the drug release from drug-eluting stents, the cylindrical forms are used. As the studies have revealed, the different types and designs of the stents such as strut number, strut thickness, angular burden can effectively influence the release behavior and make difference in the restenosis [35]. Therefore, considering the release from the cylindrical form of the stents seems to be effective when the aim of the study is to investigate the effect of the design or the performance of the final product.

As it is evident for precise investigation of a parameter it is necessary to limit the variation in the other parameters in order to obtain a consistent and clear conclusion. Therefore, recently, some studies which are focusing on the parameters, apart from the final stent design, are considering the simple geometrical model of a stent. Moreover, in this case if the other external parameters are the same, the results can be beneficial for the other kinds of the drug delivery system (DDS) such as patches, implants, micro-particles, etc. For example, in a study by Joachim Loo et al. [36], they have investigated the drug release of two kinds of hydrophobic and hydrophilic drugs from the PLGA and PLLA films with the dimension of 2 × 2 cm^2^ and the thickness of 55 µm fabricated through an irradiated-multi-layer approach. In another study by Pang et al. [37], they have studied the release of ibuprofen from the PLGA films with the dimension of about 33 cm^2^, which is as large as Baker’s dimension. Another example is the study on the drug eluting stents by Steele et al. [38], where the paclitaxel release from the PEG/PLGA films with the dimension of 3 × 1 cm^2^ and the thickness of the 15–20 µm were investigated. For investigating the effect of the shear stress on the release kinetic of the sirolimus from the PLGA, Zheng et al. [39], have used the PLGA films with the thickness of 0.11 mm. Considering film shape of the samples also facilitate the study of the characterization of the material as an example in a study of Vey et al. [40], where they have investigated the degradation in the PLGA films with different ratios of co-polymers with L/G molar compositions with the thickness of 0.3 mm in PBS solution. In another study by O’Brien et al. [41], they have studied the effect of the pulsatile flow on the fluorescein-sodium distribution from the polyurethane films with the dimensions of 0.24 × 0.35 × 1.5 mm^3^. Considering the simple rectangular shape is representative of a strut of a stent, which is a factor of the real strut dimension, Figure 5 shows some examples of utilizing different geometries of stents in the in vitro trials. This method is also utilized for determining the influence of the stent on the hemodynamic variation in blood flow. In some studies [42,43,44,45] for investigating the drug distribution around the strut of a stent, a rectangular shape was considered as a model for considering a strut of a stent. 

## 3. Factors Affecting Drug Release from Drug-Eluting Stents (In Vitro)

### 3.1. Release Compartment

As previously mentioned, the complexity of designing a universal test apparatus for stents is due to the stent’s complexity, as well as the dynamic arterial compartments of the blood and vascular wall. These are the source of numerous biological, chemical, and physical mechanisms that are more commonly observed in tissue, polymer, and lumen mediums. Figure 6 summarizes some of these mechanisms contributed at each region, and the details of these mechanisms are not in the aim of this article. Considering these mechanisms affect the release, they are a complexity of the in vitro trials.

#### 3.1.1. Artificial Blood

One of the important parameters, which has a key role in achieving the accurate drug release results from the drug-eluting stents in the in vitro condition, is the flow medium. As blood consists of plasma, red blood cells, white blood cells, and platelets, therefore seem not very easy to replace with a real representative of blood. Moreover, using real blood in the in vitro experimental is not very applicable. However, the aim of the experiment defines all the needed parameters. As the portion of the white blood cells (less than 1% of blood) and platelets (less than 1% of blood), the red blood cells (40–45% of blood) are not very high compared to the plasma, which makes up 55% of the blood, but their reaction to the drug and implanted material, and more importantly, their real impact on the purpose of the experiment should be considered.

Ignoring the probable effect of the blood cells in the therapy experiments, plasma is the essential liquid determining the blood properties. It consists of 91% water, 7% proteins, 2% nutrients, hormones, electrolytes such as sugar, lipids, insulin, sodium, potassium, and calcium. Efforts have been made to simulate the components in this order, and different types of media are considered for the substitution with biological flow. Not all the essential parameters are mimicked by these artificial solutions, but the reason for the choice is dependent on the aim of the experiment. The water, phosphate buffered saline (PBS), organic component/PBS, and inorganic component/PBS are such common solutions that have been used as a medium. In particular, the pH and viscosity of the medium are important parameters that are impacting the drug release kinetics.

In general, PBS with a pH of 7.4 has been considered in many studies as a medium for studying the drug release [46,47,48,49,50,51]. Due to the ability of PBS to keep the pH constant and the proximity of its ions to the ions of the body, it has shown a good choice in analyzing the release of the drug. However, using this medium to evaluate the release of some drugs such as sirolimus has been challenging. Studies showed that sirolimus showed very low stability in the buffer solution with pH 7.4 [52]. Where hydrolyses of sirolimus to form newer compounds with opened lactam ring at alkaline pH and buffer salts was a problem that led to its instability in these environments. Thereafter, Naseerali et al. [53], studied the influence of various media on the release profile of sirolimus from a drug-eluting stent. Their results suggested that the medium consisted of 9:1 (*v*/*v*) of normal saline and isopropanol can be considered a suitable medium for the investigation of the in vitro release kinetics of sirolimus. According to their study, obtaining a medium with the least degradation of the drug, and also an environment that provides a slower release kinetic, was considered to be an essential parameter in order to achieve the suitable environment.

Pruessmann et al. [34], studied the impact of deionized water, PBS and phosphate-buffered saline without sodium chloride (PB) as a medium on the kinetic release of triamterene from coated stents. According to their study, deionized water showed the greatest release compared to PBS and PB. This increase was justified due to the higher solubility of the triamterene in deionized water and PB compared to PBS. This shows the important effect of increasing the soluble polarity by sodium chloride on organic compounds in aqueous solutions, which in turn leads to a decrease in the solubility of non-polar solutes such as triamterene. It is worth noting that according to the results of this study, the properties of the drug should always be considered as an important parameter in order to select the medium as it clearly has an effect on the kinetics of drug release, and besides, the common properties of the artificial media and real media in the body should be preserved.

As mentioned, the type of drug should be considered as an effective parameter in choosing the media. In particular, the solving of problems such as solubility and stability of non-polar drugs that have shown low solubility in an aqueous medium has attracted the attention of many studies. In fact, the ‘-olimus’ groups of drugs are not stable at alkaline or neutral pH. So, lowering the pH to less than 7.4 could be a way to reduce the degradation of these drugs in the media. Furthermore, improving the solubility of the poorly water-soluble drugs in the aqueous media by using suitable solvents, such as surfactants, have been one of the suitable approaches that have been used in this field. If sirolimus (SRL) release from DES was taken as an example in this regard, Raval et al. [54], worked on optimization of media for the release of SRL from drug-eluting stents. They buffered the media at pH 4 for minimizing the degradation of SRL. In order to increase the solubility of the drug, a special amount of surfactant was used. Their results proposed that a release medium consisting of 0.1% P123 (kind of PEO–PPO–PEO block copolymers) in phosphate buffer pH 4.0 was most suitable for in vitro release of it from DES. In another study by Jelonek et al. [55], they have used acetonitrile and methylene chloride as a media for analyzing the SRL loaded in PLLA and PDLA. It is worth noting that the solutions, which were used as media, were the solvent for solving both the drug and the polymeric carrier. This solution is used, rather, to extract and quantify the entrapped drug (SRL) in the DES.

It should be noted that the use of solvents and surfactants could increase the kinetics of drug release from stents. As in most cases the drug released from the stent is loaded on the stent by being trapped in the polymer coating, the use of solvents and surfactants can increase the rate of degradation of the polymer coating. On the other hand, the use of surfactants can affect the separation and quantification of SRL by HPLC, which can lead to erroneous results.

In another study by Pruessmann et al. [34], they have analyzed the effect of different release medium, PBS and deionized water, on the release of the triamterene from the DES. The results indicated that triamterene has been released about two times higher in deionized water compared to PBS. 

In another study for analyzing the effect of the struts and also distribution of the drug in the two media (artificial blood and artificial tissue) the viscosity of the medium was amongst the important parameters to be considered, therefore a mixture of glycerol-water (40/60 vol%, 0.01% surfactant) with the µ = 0.0044 Pa.s, and ρ = 1101 kg/m^3^ resulting in a kinematic viscosity of 0.04 cm^2^ was used, which is similar to that of blood [41,43].

Merciadez and his colleagues [56] have used a new medium containing an organic solvent prepared using 2% ultra-pure sodium dodecyl sulfate (SDS), in high purity water with 10% gradient-grade acetonitrile (ACN), and buffered to pH 4.5 with phosphate. The mobile phase was a mixture of 55:45:0.02 water/tetrahydrofuran (THF)/formic acid (*v*/*v*). This allowed them to correlate in vitro release profile with the in vivo one from the porcine.

Moreover in a study by Chabi et al. [45] where the hemodynamic of the flow with the presence of stent was under consideration a mixture of 87% of glycerol and 13% of water was used.

#### 3.1.2. Artificial Tissue

The release of drug from DES is only one part of the story. On the other side the amount of the drug diffused to the artery is the main clinical goal and certainly more difficult to investigate. Clinicians advise that a uniform drug concentration should be attained across the arterial wall, and the concentration should be maintained within some therapeutic window [57]. In this regard, studies are trying to measure the amount of drug that has penetrated the vessel by using different gels that can simulate the vessel artery.

Neubert et al. [58] applied the calcium alginate hydrogel for simulation of the vessel artery. The water content of the gel was approximately 96%. Stability at 37 °C, the feasibility to adapt gel strength and elasticity and the mild gelling conditions, which allow for the incorporation of diverse substances such as proteins or living cells, were the reasons for choosing the calcium alginate hydrogel matrix. The gelling time can also be adapted by variation in concentrations and the gelling mechanism, which can directly affect the drug diffusion in the gel. Furthermore, the results of this study showed that the presence of another compartment (hydrogel) changed the kinetic of the release from the samples, which was accompanied by a decrease in the amount of drug released into the flow medium. In addition, according to the results, the current composition of the medium and the hydrogel was not representative of the in vivo condition, and it needs more adaptations of the hydrogel compartment or periodic medium replenishment.

Depending on the environmental conditions of setup for analyzing drug released from the stents, some properties of hydrogels such as rheological properties, degradation and swelling are among the factors that will affect their selection. In a study by Semmling et al. [59] the hydrogels of 2 wt.% agar, 2 wt.% agarose, 10 wt.% PAA and 15 wt.% PVA were selected. In order to find a measure for the long-term stability of the gels, the mechanical properties of the prepared gels were determined by texture analysis. In this regard, stress–strain curves of native gels, as well as gels that had been perfused with phosphate-buffered saline (PBS) pH 7.4 in their setup for 28 days were studied. Their results showed that agarose gel seems to be the most suitable candidate for long-term dissolution testing since the target gel parameters are relevant for their use as a tissue simulating compartment in their setup.

It should be noted that in addition to the desired physical and rheological properties, in order to select a hydrogel as a vessel wall, the drug penetration coefficient in the hydrogel has great importance and should be considered as a parameter with high status. In a study by Semmling et al. [34], the penetration coefficient of triamterene was under 3 wt.% alginate, 2 wt.% agar, 2 wt.% agarose, 10 wt.% PAA and 15 wt.% PVA hydrogels were examined (shown in Figure 7). According to their results, the penetration coefficients of these gels were in the range of 2 × 10^−4^ mm^2^/s (PVA) to 8 × 10^−4^ mm^2^/s (Agaros). The results of these studies showed that the hydrogel with the lowest diffusion coefficient had 4% of drug penetration. This is where the penetration coefficient of sirolimus in the human coronary arteries is reported as 1.5–2.5 × 10^−4^ mm^2^/s [60], which is near to the penetration coefficient of triamterene in hydrogels used in these experiments.

Bandomir et al. [61] studied the amount of paclitaxel diffused in the calcium alginate, polyacrylamide (PAAm) and poly(vinylethylimidazolium bromide) hydrogels from a drug-coated balloon. Certain properties such as permeability, flexibility and long-term stability of synthetic hydrogel, were the properties which are considered for introducing a good candidate as the vessel wall. The dissolution of the network by monovalent cations such as Na^+^, as well as its susceptibility to microbial contamination, were among the disadvantages mentioned for calcium alginate hydrogel. 

In order to select the type of hydrogel and to obtain and select the suitable cross-linkers used to make the hydrogel, it is important to know the mechanism of drug transport to the hydrogel. Studies have shown that drug delivery from drug-impregnated stents is controlled by both penetration and convection mechanisms. Given that the penetration mechanism will occur based on the concentration gradient, firstly, the drug must be dissolved in the matrix and subsequently, the drug can penetrate to the gel or media [62]. For this purpose, the solubility of the drug in the matrix, and then the penetration of the hydrogel, should be considered as the important parameters. For example, a study of a balloon impregnated with paclitaxel in a hydrogel showed that due to the low solubility of the paclitaxel in the aqueous medium, the amount of drug transferred to the hydrogel was by mechanical forces during balloon expansion (shown in Figure 8) [61]. In addition, the research has shown that biological reactions, such as binding the drug particles with the receptors in the tissue, affects the drug penetration in the tissue for simulating in vivo reactions, and some proteins were added to the hydrogel. The presence of proteins can cause the drug to deposit in the vessel wall, which after a while can allow the drug to penetrate into the tissue [63]. Therefore, it is worth noting that when the drug has low solubility, after opening the stent by balloon, part of the drug can be transferred into the gel by mechanical force, but after that, the remaining amount of the drug, due to the inability to penetrate into the gel, will only wash-off from the stent and this can cause an error in calculating the amount of drug that has penetrated into the gel.

Semmling et al. [64] examined the effect of using different hydrophobic additives to the vessel wall simulating the hydrogel compartment on release and distribution from model substance-coated stents. In this regard, four alginate-based gel formulations containing reversed-phase column microparticles LiChroprep^®^ RP-18 or medium-chain triglycerides in the form of preprocessed oil-in-water emulsions Lipofundin^®^ MCT with different concentrations were chosen. In general, use of additives was applied for improving the media contact with the hydrogels used. It is worth mentioning that in this study, fluorescein and triamterene were studied as hydrophilic and hydrophobic drug models, respectively. The results showed that the effect of gel improvement had no significant effect on the penetration of the hydrophilic drug into the hydrogel, while the improved gels were a more suitable substrate for the transfer of hydrophobic drug into the hydrogel. As an example, Figure 9 shows the comparison of amount of the drug detected in the media, in the hydrogel and the residual drug in the polymer with considering alginate gel versus LiChroprep gel containing 5% *w*/*w* LiChroprep^®^ RP-18.

Another important and effective factor with hydrogels for the penetration of drugs through them, is the suitable choice of the base agent within the hydrogel. Pruessmann et al. [34] investigated the effect of deionized water, PBS and PB as the base for preparing hydrogels for the diffusion of triamterene in them. Their study showed that more drug was transferred to deionized water-based hydrogels than PBS and PB-based hydrogels. This effect was due to the absence of salt in deionized water, which was discussed in the previous section. Table 1 shows a summary of the comparison of the artificial blood and tissue used in the in vitro studies.

In the experimental tests considering the tissue layer, whether ex vivo, in vivo or in vitro, it should also be considered that stent positioning on the arterial tissue is very important, where malappositioning of stents highly affects the release results [65,66].

### 3.2. Release Test Methods

Analysis of the kinetics and amount of drug released from the drug-eluting stents play the most important role in evaluating a drug-impregnated stent. Due to the cost and time-consuming in vivo tests, examination by in vitro tests has been the center of attention in many studies. Studies have always been trying to increase the accuracy of the in vitro tests in order to provide a good estimate of the results in comparison to in vivo tests. In general, the methods used for this purpose are divided into two static and dynamic conditions. The main difference between the two tests was the use of media flow. However, the temperature used in these two methods is 37 °C and has been accepted by researchers.

#### 3.2.1. Static Condition

The static method has been used as a common method in measuring drug release from drug-impregnated stents and other drug delivery systems. In general, in this method, the stent is immersed in a certain amount of media, and then the sampling of the media will be completed in certain periods of time. The drug release mechanisms in this method occur mainly through the concentration gradient between the DES and the media. The test temperature will be constant during the drug release test.

One of the important points for testing the drug release in the static state is media sampling. In this way, at the time of sampling, what volume of media should be replaced with fresh media [67]. In the study by Khan et al. [68] the rapamycin release from drug-eluting stents was evaluated under static conditions. In this study, the sample was immersed in 2 mL of medium. In order to evaluate the amount of drug released, 1.5 mL of medium was replaced with fresh medium at each time point. The sampling time for the experiments was six hours, one day, three days, five days, ten days, and then weekly up to seventy-five days.

As mentioned, the evaluation of drug release from stents is very important to determine the effectiveness of stents, and studies have always tried to bring the test conditions closer to the body environment. Abbasnezhad et al. [69] showed that there is a significant difference between the drug release under the static and dynamic conditions (the next section will discuss more on the effect of flow on drug release). One of the drawbacks of the static method is the absence of circulating flow. Moreover, since the release of the drug in the static state will have slower kinetics than in the dynamic state [70], and more experimental time will be needed. However, due to the availability of this method, it seems that useful basic information can be obtained from this test.

#### 3.2.2. Dynamic Condition

As already shown in Figure 6 several drug release mechanisms contribute to the drug release from the drug-eluting stents. The most important mechanisms consists of the diffusion-controlled drug release, dissolution/degradation-controlled drug release, and osmosis-based controlled release [71]. It is worth noting that each mechanism provides a different kinetic of the release from the stents. A study by Abbasnezhad et al. [72] has showed the effect of the flow rate on the kinetic of the release by considering the associated mechanisms.

The use of shakers has been amongst the common methods of measuring the amount of drug release at the dynamic condition. For this purpose, a certain number of stents (usually one stent) to media are placed in the screwed-glass vials [73], tubes [74], or flasks [75], and then the test is performed at a certain agitation of the shaker. In most studies, the temperature intended for the experiment was kept constant at 37 °C by an incubator [72,76] or a water bath [73,77]. The agitation speed of shaker in studies considered in the range of 50 [78,79], 75 [80,81], 80 [75], 100 [73,76], 120 [77], 130 [82], 175 [74], 250 and 300 [83] rpm. Sampling in this method to check the amount of drug released has been one of the variable parameters, which consist of several methods. Generally, the methods of sampling are as follows:Changing the special volume of media with fresh media (fresh media is added to keep the test volume constant and avoid the saturation) [83];The media is completely replaced with new media at specific times [73,75,76];A specific portion of the media is removed to analyze and returned to the test environment after analysis [81].

To the best of our knowledge, the effect of shaker agitation on drug release has not been studied by simulation. In addition, in the studies, the reason for choosing a special agitation to evaluate drug release was not specifically mentioned, and this value was in the range of 50 to 300 rpm. In general, it can be said that the conditions created by using shakers, due to the importance of the presence of flow on the drug release, cannot give an accurate estimate of drug release of the in vivo tests, but this method will accelerate the drug release from the stent and can reduce the time required to perform the test.

Apart from the shakers there is another method for the dynamic condition, which is the circulation of the flow at the continuous state, and this condition may have more similarities to the real case comparing to the shakers. The same can be said for agitation for the continuous state, which also has different values for the flow rate that has been chosen in different studies. 

As an example in a study by Bandomir et al. [61], a flow rate of 35 mL/min was chosen for studying the drug delivery from the drug-coated balloon, whereas, in a study by Zheng et al. [39] the flow rates of 3, 10, 30 mL/s were chosen for the Sirolimus release from DES.

Another study for the release of Sirolimus from DES has been conducted by Merciadez et al. [56], where in this method a medium with the flow rate of 25 mL/min was used. This value was chosen due to the optimum pump performance with respect to priming and flow characteristics of the Sotax apparatus media pump.

Another study [84] has considered the laminar flow rates of 6.8, 10, 11.6, 12.3 and 17.3 L/min where they have selected the different laminar flow rates of an oscillated heart pulse.

In an investigation by Seidlitz et al. [85] they have used the flow rate of 35 mL/min, where they have referenced the flow rate in the coronary vessels, as well as the flow rate of 4 mL/s for two different types of drugs. Their analysis indicated that the variation in the flow rate had not a distinct effect on release and distribution. Therefore, they have concluded that the effect of the flow rate should be analyzed by case-by-case examinations through individual assessments of their sensitivity to such changes. 

In another study by Bernard et al. [86] they have considered two flow rates of 60 and 140 mL/min as the minimum and maximum values of the right coronary artery. They have stated that an increase in the flow rate emphasizes fluid perturbations, and generates a wall shear stress rise except for inter-strut area.

Moreover, to the continuous flow in the case of dynamic condition, pulsatile flow and especially systolic-diastolic flow pattern are in the interest of studies, which can simulate the real flow pattern of the blood circulation in the body.

However, in some studies such as Vijayaratnam et al. [42,43] they have performed experimental and CFD simulations to investigate the impact of luminal blood flow patterns on the drug transport behavior of stented arteries and have concluded that neither the pulsatility of the flow, nor the viscosity of the flow change the results of the drug uptake, Figure 10 shows the experimental test rig and the type of the flow used in this study.

In another study [87] the comparison of the continuous flow and pulsatile flow, experiments have shown that the pulsatile flow shows a different character of flow, especially at the bifurcations, where the steady flow shows a fixed separation of the flow, without any turbulence and wall shear stress, which is rather constant during the time. However, the pulsatile flow does not have the constant region of separation, with the possible turbulence at the end of the systole and the wall shear stress that varies in magnitude and direction and, moreover, increases the time of the resistance of the bubbles.

### 3.3. Apparatus for Release Testing

In vitro drug release from the DES is a big challenge for researchers. In this regard for designing an apparatus, which can be specialized to contain the similar condition of the in vivo condition such as treatment of the vessel wall in the vicinity of the DES and the lumen side, providing a systalic-diastolic flow pattern also improvising a system for filtering the drug released in the medium are amongst the new plans for improving the release test setups. In the following the most famous apparatus, which are commonly used, will be discussed.

In the static state generally, laboratory vials are used. However, in the dynamic state shakers, basket apparatus USP 1 (United States Pharmacopeia), or paddle apparatus USP 2, which are normally destined for capsules and tablets, have only one difference between them, which is the replacement of the paddle to the basket for drug carriers that float or drop at the bottom of the vessel. Reciprocating Cylinder USP 3, which was developed to mimic gastrointestinal test, flow-through cell apparatus (USP 4), which has a continuous flow circulation it is designed for low soluble drugs, implants and suppositories. They can be used in two types of open type or closed-loop system, where the pump delivery in this case is about 240–960 mL/h. Compendial Apparatus 5 (paddle over disc) is similar to the paddle system (USP 2) but with an additional disc mounted on it.

Cylinder type or USP 6, resembles the basket type but the basket and shaft are replaced with a cylinder stirring element. USP 5 and USP 6 are normally destined for drug release from the transdermal patches. 

Reciprocating holder apparatus with agitation (USP 7), this type is the most recent apparatus destined for different types of drug carriers such as tablets, capsules, transdermals, osmotic pumps, and arterial stents, with different agitation speeds. 

Generally in the case of stenting, these devices are not sophisticated enough [69,88]. The importance of the systolic-diastolic flow and the inherent pressure variation is indicated in in silico models, which are parameters that are not really respected in the experimental tests [89]. The presence of a simulating arterial tissue is another important element affecting the release, where the efforts in this regard lead to the development of the Vftc (vessel-simulating flow-through cell) method [90]. However, so far, the simulating tissue is not accurately reproducing the characteristics of the real one [91,92,93]. The apparatus USP 2, USP 4, USP 7 and vFTC in the closed-loop mode, are amongst those that have taken attention away from the DESs in the field of drug delivery.

For comparing the effect of the different test setups on the release profile, a study by Medina et al. [94] has compared the release profile of the ibuprofen obtained by two methods of USP 2 and USP 4 with the reference by model-independent, model-dependent, and analysis of variance (ANOVA). Their results have indicated that the release profile obtained by the method of USP 4 was similar to the profile of the reference.

In another study Pruessmann et al. [59] have studied the release of triamterene as a model substance from the DES at three different test setups: USP apparatus 7, USP apparatus 4 (FTC) and vessel-simulating flow-through cell (vFTC) (shown in Figure 11). According to the results obtained in in vitro experimentation, they have stated that dissolution vessel geometry and medium volume had no influence on the release behaviour, whereas the flow through cell method had a lower release rate than the incubation methods.

The same method of vFTC was used by Seidlitz et al. [85] for the comparison of the dissolution and release results of fluorescein sodium and triamterene from stent coatings by the methods of the vessel-simulating flow-through cell with a standard paddle (USP 2) and flow-through cell (USP 4) apparatus. The results showed that release from the coating was decelerated by embedding in the hydrogel in the adapted apparatus, shown in Figure 12. However, in another study by Pruessmann et al. [34] triamterene showed a higher release in the medium at the first stage by vFTC method, then it is underneath of the profile of the release obtained by FTC method. 

The studies in this regard show that to achieve an apparatus, highly adapted for studying the drug release from drug-eluting stents, there is already much to be done. Moreover, in vitro/in vivo correlations, by considering the different parameters, in order to personalize the therapy and thus increase the efficiency of the therapy, remain a challenging topic.

### 3.4. Analytical Tools to Determine Drug Release

The first step to determine the efficiency of a drug delivery carrier is to evaluate the amount of drug released from the carrier at different times. The main technologies used in this aim are: High-performance liquid chromatography, UV-Vis spectroscopy and Raman spectroscopy.

#### 3.4.1. High-Performance Liquid Chromatography

Liquid chromatography is a classical technique used to separate a sample into its individual parts. This separation occurs based on the chemical or physical interactions of the sample with the mobile and stationary phases. There are many different types of chromatography techniques and systems available for a wide range of applications, all of which are defined as High Performance Liquid Chromatography (HPLC). HPLC analysis focuses on macromolecule isolation through chemical interaction, affinity or hydrodynamic volume. As the molecules present in the mobile liquid phase interact differently with the stationary phase, they produce different signals on the detectors. This technique is generally associated with other analytical techniques (optical detectors), such as UV detectors or fluorescence detectors.

#### 3.4.2. UV-Vis-Detector

Ultraviolet-visible (UV-Vis) spectrophotometry is a technique based on the absorption of photons belonging to the UV, visible or near IR regions of the electromagnetic spectrum. Substances exposed to such radiations are prone to electronic transitions. The signal consists in a series of peaks in an absorbance versus wavelength spectrum. Compared to HPLC, UV-spectrophotometry method is much faster and less expansive.

By this approach, aromatic structures, unsaturated (conjugated) compounds, or carbonyl groups inside the molecule are detectable. Compared to HPLC, UV-spectrophotometry method is much faster. Determining of solution using HPLC-UV needs organic solvent (HPLC grad) as flushing and mobile phase. Moreover, HPLC is more expensive than the UV-spectrophotometer.

In a study by Dhole et al. [95] they have compared the two methods of HPLC and UV for the detection of Repaglinide drug in the tablets. They have stated that two methods have enough accuracy, however, HPLC is prone to being more precise. The approach of UV compared to HPLC has the benefit of not requiring the time-consuming treatment and processes associated with chromatographic methods.

#### 3.4.3. Fluorescence Detector

Fluorescence detectors are probably the most sensitive among the existing modern HPLC detectors. It is possible to detect even a presence of a single analyte molecule in the flow cell. Typically, fluorescence sensitivity is 10–1000 times higher than that of the UV detector for strong UV absorbing materials. Fluorescence detectors are very specific and selective among the other optical detectors. 

Compounds having specific functional groups are excited by shorter wavelength energy and emit higher wavelength radiation, which is called fluorescence.

Fluorescence detector is a method where the compounds often contain conjugated aromatic ring systems, can be detected via it. Due to a missing chromophore, the study of amino acids can be very difficult, so neither UV-Vis nor fluorescence detection can be used for detection. 

#### 3.4.4. Raman Spectroscopy

This technique consists in illuminating the sample with monochromatic light (usually from a laser in the visible, near infrared, or near ultraviolet range, although X-rays can also be used). The interaction of this light with the molecules of the sample depends on their vibrations (this phenomenon is called inelastic diffusion or Raman scattering) and a shift in the energy spectrum of the diffused photons is observed. This method is more advanced compared to UV and HPLC. One of its advantages is the ability to use it in situ during the experimentation, therefore it decreases the use of disposables, it is fast, precise and has less risk due to less transportation of test substances [96,97]. In a study by Bourget et al. [97] on the detection of an anti-cancer drug, the authors have pointed out that both approaches offer sufficient precision and accuracy. Raman spectroscopy, on the other hand, is beneficial for ensuring the analytical quality of this drug and helps to safeguard careers and their working environment.

### 3.5. In Vitro–In Vivo Correlations

The importance and evaluation of in vitro tests are further emphasized when they are representative of an in vivo test or are used to predict the in vivo performance of the active substances. Therefore, the attempts are for developing a correlation between in vitro and in vivo tests, which presents the major challenge of the coming decades. IVIVC is described by the US FDA as a predictive mathematical model that describes the relationship between an in vitro property of an extended release profile (usually the rate or extent of in vitro drug release) and a related in vivo response, for example, plasma drug concentration or amount of drug absorbed [98].

This relationship may be qualitative or semi-quantitative, such as the correlation of the mechanism of release. This method is more developed and used for the oral administration, where it permits the development of drug administration and controls the dosage between the therapy window.

In this case important parameter between the in vitro and in vivo test methods is the difference of the physiological properties of the plasma with in vitro medium and the absorption of the drug in the biological environment. A mathematical model for the correlation between the release mechanisms and also a correlation between the entire release time of the in vitro and in vivo methods would be most helpful and most effective IVIV correlation [99].

Therefore, a validated predictive IVIVC will have a substantial positive effect on the consistency of the commodity, the productivity of production and decreased compliance costs.

For developing the IVIVC model, both in vitro and in vivo results are requested, where the results could be different for a certain logic with a certain difference in the release rate. 

According to the statement of Qiu and Duan [100] for developing the IVIVC mathematical model, relating the entire release of the drug in vitro and predicting the drug absorbed or the concentration of the drug in plasma, the mathematical correlation is applied in a two-stage procedure. Figure 13 shows this procedure.

Developing an IVIVC model needs in vitro drug release data at different release rates (e.g., fast, medium, and slow) and a discriminating in vitro test methodology [100].

Different types of the mathematical models and methods are proposed for developing the correlation IVIV. A single stage model compares the in vitro release with the in vivo drug concentration in the plasma, where drug concentration degrades in time, therefore the parameters such as C_max_ and T_max_ are notable. The second type is two-stage correlation, which is more straightforward as the released drug and absorbed drug, respectively, in the case of in vitro and in vivo, are compared. An example of a two-stage approach is described in Equation (1):(1)$${\left(\%absorbed\right)}_{in\;vivo}=a+b\;{\left(\%released\right)}_{in\;vitro}$$
where, it comprises the lag and differences of the kinetic, in vivo, and in vitro tests.

It is notable that considering all the variable parameters may be difficult in identifying the effect of that parameter on the model, therefore a distinct variable is needed for emphasizing the effect of each parameter in the model.

For the single-stage approach, a model in this regard can be a poly exponential model, as Equation (2).
(2)$$C_{\delta }\left(t\right)=\sum_{i=1}^{nex}A_{i}\times e^{-\alpha_{i}\left(t-t_{lag}\right)}$$
where, nex is the number of exponential terms in the model, $t_{lag}$ is the absorption lag time, and $C_{t}$ is the plasma concentration at time *t*.

The type of the mathematical model, as logarithmic, sigmoidal, power, etc., can be estimated by the pattern of the release in the in vitro, in vivo cases. These patterns of the release are an indicator of the kinetic of the release where generally in most of the cases the release kinetic can be defined by one or some of the release mechanisms: zero-order, first-order, square root time of Higuchi, and exponential Peppas model.

By comparing the predicted and experimental results, the efficiency of the IVIVC is evaluated. It should be noted that the model’s ability to predict changes in in vivo plasma concentrations with differing release rates should be tested by fitting data from multiple formulations. A $C_{\delta }\left(t\right)$ function can be specified reliably by doing so. The use of at least two extended-release formulations with varying release rates in the evaluation of IVIVC is therefore one of the most important criteria of this strategy. An IVIVC model was developed for the oral delivery of components of galantamine by Jacobs et al. [101] by combining the immediate-release and extended-release and integrating the pharmacokinetic profiles of them by using a one-stage process.

Indeed, several failures in attempts to achieve IVIVC for drug administration can typically be due to the absence of an in vitro predictive test, or to a discrepancy between the formulations of the in vitro, and in vitro tests. It is therefore important to consider how in vitro and in vivo results can be influenced by the physicochemical and biopharmaceutical properties of the drug, administration technology, and formulation of a drug and its interactions with the in vivo and in vitro environment. The typical efforts in this regard were for the oral administration by changing the release media, hydrodynamics and generating shear force, which was almost at a static state that cannot totally be representative of in vivo conditions [100].

In this regard, a qualitative and quantitative relationship should be established between the drug release profiles, considering different parameters of in vitro tests. The latter will help to better choose the in vitro experimental methods and conditions in order to better correlate with the kinetic and the mechanisms of the in vivo release. 

In another study by Ma et al. [81] they have investigated the release profile of the combinations of two drugs of paclitaxel/sirolimus in vitro (measuring the drug content in PBS) and for the in vivo a rat aorta model was utilized. The results from these experiments showed that both drugs at both conditions of in vitro and in vivo had two-phase release profiles. The results are shown in Figure 14.

On the condition that the in vitro results are not correlative with in vivo results, the in vitro conditions should be improved to get closer to the in vivo condition, and therefore to the in vivo results.

In this regard, a study by Merciadez et al. [56] has tried different parameters of the in vitro release tests from DES in order to bring the in vitro release profile close to the in vivo release profile. In this regard, they have obtained a release profile of in vitro from USP apparatus 4 in 24 h, which corresponds to the release profile of 30 days in vivo in porcine. Figure 15 shows this correlation.

The parameters chosen for the in vitro test were as follows: test apparatus (USP 4); elution medium was an organic solvent; flow circulation of 25 mL/min; pH about 4.5; and temperature of 37 °C.

In another study by Sako et al. [102] they have evaluated the release of acetaminophen from the different compositions of a hydrophilic matrix, where the in vitro tests showed the same profile of the release for different agitation, however, it was not corresponding with the in vivo tests, thereafter by modifying the in vitro test conditions, the results were consistent with the in vivo results.

Some other variables are formulation design, dosage, drug properties, release environment. It is notable that different formulation of the drug carriers may have different sensitivity to each of these variables [103,104]. Therefore, it is essential to examine its sensibilities to different variables in the in vitro case before defining the IVIVC model.

One of the difficulties in this method is that each IVIVC is valid just for a specific type of dosage and drug delivery system, therefore by changing the type of the drug delivery system (DDS) or even with the same dosage and DDS, just with or without excipients can have different release mechanisms (e.g., diffusion, degradation, osmosis) where it is necessary to develop a new IVIVC [105,106].

The state of the art has indicated that there is not a universal method of in vitro test that can mimic the complexity of the test conditions of the in vivo case, for better predicting the in vivo release profile. Therefore developing an IVIVC should be studied case by case, referring to the different types of the DDS and considering different effective parameters, and their adaptability to the in vitro conditions.

For example in a study by Kim et al. [107], they have studied the durability of a polyurethane-covered stent for the biliary endoprosthesis, destined for patients with biliary stricture, and in this regard they have used the bile juice as the circulating medium. Whereas it is not applicable when the in vitro medium is used for simulating the plasma and blood components.

As mentioned depending on the type of DDS, adaptability of the in vitro conditions to the in vivo would be more or less complicated. In the case of drug eluting stents, there is high enough parameters for adaptation. In this regard, some researchers focus on one parameter, such as the strut thickness, dosage, type of the drug or type of the material, etc., whereas some others focus on multiple parameters. In a study by McGinty et al. [108] they have coated the stents with low and high dose of the Sirolimus/polymer (25/75 and 75/25) and in the in vitro case they have immersed the stents in the release medium of PBS. For the case of in vivo, they have used the porcine arterial tissue and the release profile was cumulative of the drug transport by the mechanisms of diffusion, advection and binding. Their investigation showed that the discrepancy for the results of high dose DES was notable, whereas the results for low dose DES were more consistent (shown in Figure 16). This may show the complexity of high dose DDS, where there may be other mechanisms more than diffusion–dissolution to be considered.

## 4. Study Limitations

Some of the limitations of developing the in vitro test bench for simulating the real condition of the drug release from the drug-eluting stents can be as follows: 

Considering properly the two contacting environments with DES (lumen side, blood, and the abluminal side, tissue compartment); considering all the physical, chemical, and biological mechanisms that are happening consequently or simultaneously in the release system, an example is the reaction of the cells to foreign objects such as the inflammation response, which is complicated to be considered in the in vitro case; considering the aspect of filtration in closed-loop systems; a concrete method of sampling in the static and continuous flow. It is notable also that binding proteins specific to each type of the drug, which can be placed in the plasma, intima, media, or adventitia will change the drug distribution in the artery. Moreover, malpositioning the DES can strongly affect the release results, and on top of that, the topology of the remained lipids will also be among the influencing parameters, which are challenging to simulate.

It is worth mentioning that it would be difficult for a biorelevant test system to accurately reproduce all in vivo conditions since the in vivo situation is highly complicated and prone to be rather different among different patients and different zones of diseases.

## 5. Conclusions

Interests in universal apparatus for in vitro drug release from stents are growing. In this regard, the main challenge is to simulate the in vitro conditions as much as possible in the in vivo case. The final goal is to develop a model of correlation where the in vitro tests can predict the release profile in the in vivo case. Furthermore, we wish to see the development of a test facility that can provide the best possible bio-relevancy and adequately help researchers to develop the IVIVC most effectively, as this has not yet been sufficiently developed in the case of drug-eluting stents. The IVIVC developed so far is mostly applied to the oral administration, and the complexity of regenerating the in vivo condition in the in vitro tests has retarded the advancement of the IVIVC in the case of the DES.

Among the fluid media used as a replacement for the blood, the biological and biochemical aspects have not been considered. The same applies to hydrogels, where the bio-relevance parameters such as the consideration of receptors for binding, consideration of the consequence of tissue damage following DES deposition, inflammation, etc., are not sufficiently considered in the current research. However, hydrogels continue to be the simplistic candidate for mimicking the tissue arterial, considering the mechanical stability, diffusion coefficient, rheological behavior, and swelling.

The current bio-relevant apparatus has been designed to mimic some aspects of the in vivo situation, which may influence the drug distribution. The developed apparatus will facilitate the advancement of vascular implants for personalized therapy. This is a substantial challenge, however, it is one with undeniable benefits for patients.

## Figures and Tables

**Figure 1 polymers-14-02751-f001:**
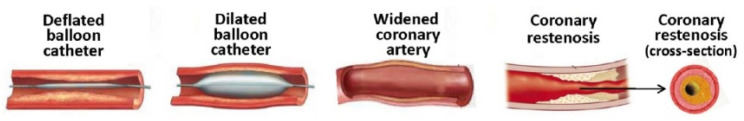
Coronary restenosis after coronary angioplasty with the balloon, Reproduced with permission from [3].

**Figure 2 polymers-14-02751-f002:**
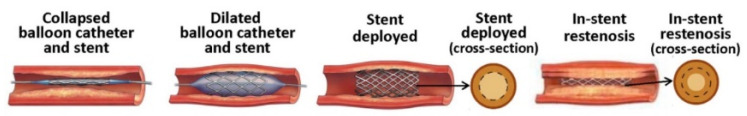
In-stents coronary restenosis after coronary angioplasty with the bare metal stents, reproduced with permission from [3].

**Figure 3 polymers-14-02751-f003:**
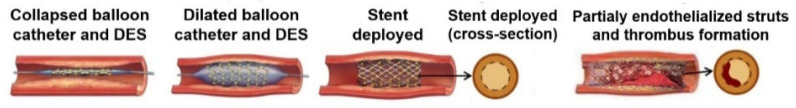
Late thrombosis after drug-eluting stents, reproduced with permission from and modified from [3].

**Figure 4 polymers-14-02751-f004:**
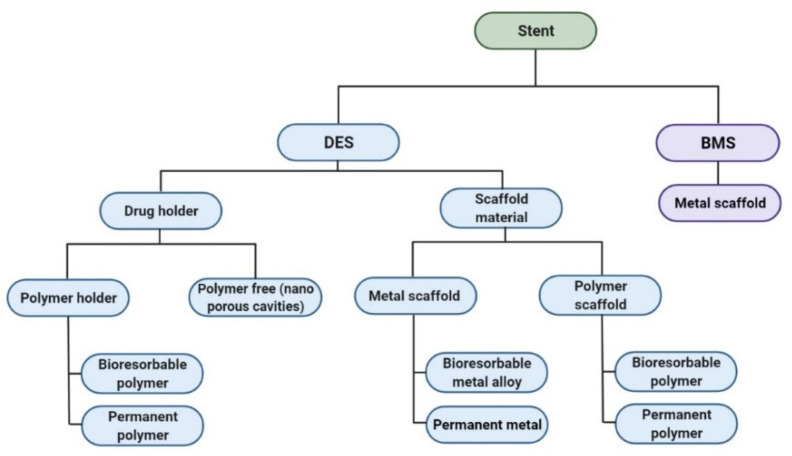
Schematic of the material choice in the various stents [19,23,24,25,26,27,28,29,30].

**Figure 5 polymers-14-02751-f005:**
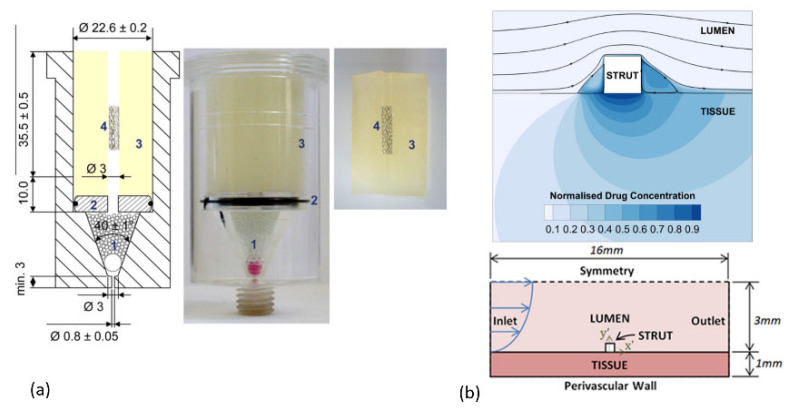
Different geometries of stents (**a**) cylindrical (1. Glass beads, 2. acrylic glass disc, 3. hydrogel and 4. expanded stent) and (**b**) rectangular used in different studies, reproduced with permission from [46,47]. Reproduced with permission from Anne Seidlitz, Stefan Nagel, Beatrice Semmling, Niels Grabow, Heiner Martin, Volkmar Senz, Claus Harder, Katrin, Sternberg, Klaus-Peter Schmitz, Heyo K. Kroemer, Werner Weitschies, Examination of drug release and distribution from drug-elutingstents with a vessel-simulating flow-through cell; published by Elsevier, 2011.

**Figure 6 polymers-14-02751-f006:**
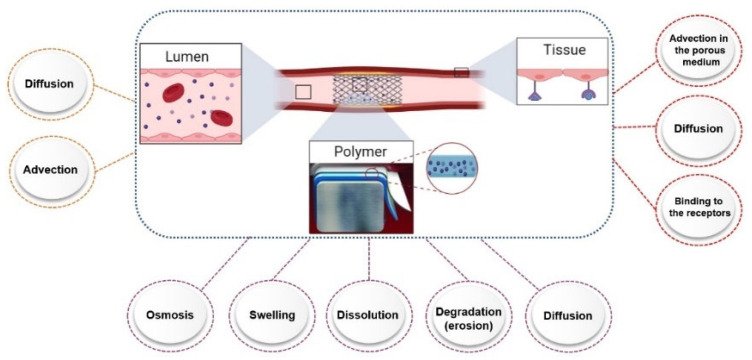
Contribution of different mechanisms at different compartments, affecting the drug release from DES.

**Figure 7 polymers-14-02751-f007:**
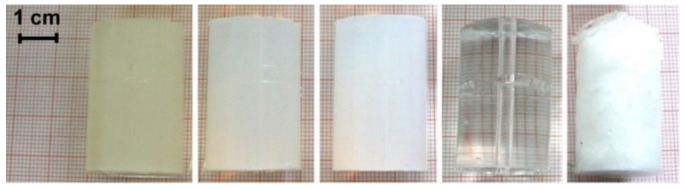
Photographs of freshly prepared native hydrogels of the final gel formulations: 3 wt.% alginate, 2 wt.% agar, 2 wt.% agarose, 10 wt.% PAA and 15 wt.% PVA (from left to right), reproduced with permission from [59].

**Figure 8 polymers-14-02751-f008:**
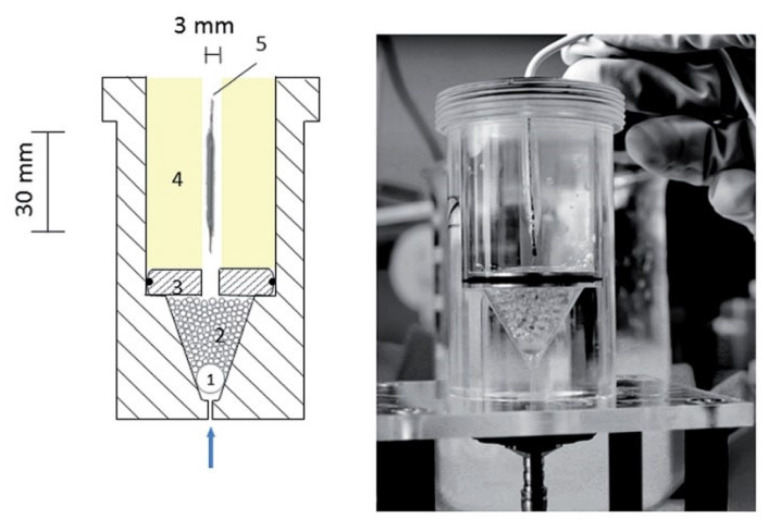
Schematic drawing of the flow through cell: 1 large glass bead; 2 small glass beads; 3 metal disc; 4 hydrogel matrix; 5 drug eluting balloon (DEB), reprinted from [61].

**Figure 9 polymers-14-02751-f009:**
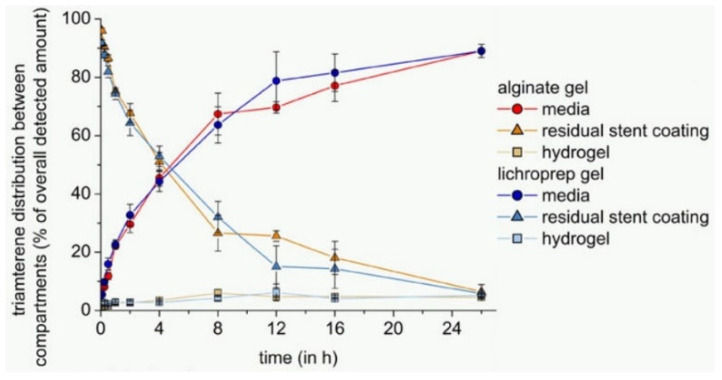
Comparison of reference and modified hydrogels. Alginate gel versus LiChroprep gel containing 5% *w*/*w* LiChroprep^®^ RP-18. Cumulative amounts (%) of triamterene detected in PBS pH 7.4, respective gel formulations, and residual model substance fractions within the stent coating, reprinted from [64].

**Figure 10 polymers-14-02751-f010:**
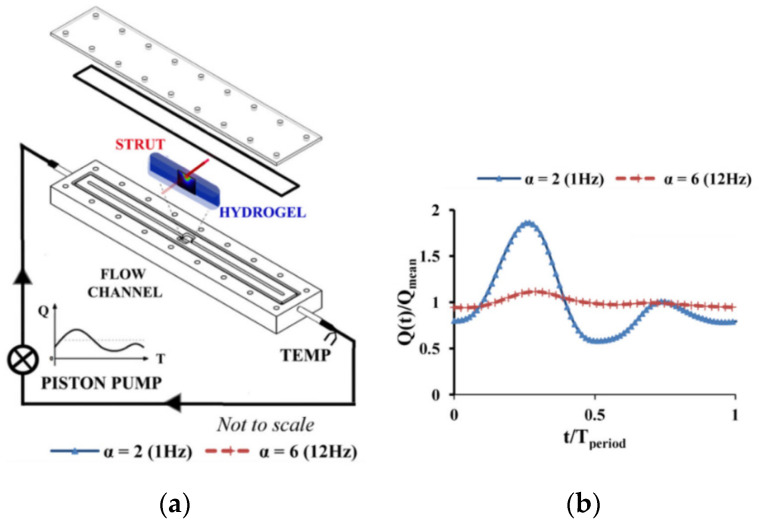
(**a**) Experimental test bench, (**b**) pulsatile and steady inlet flow rate waveforms used in this study, reprinted from [42].

**Figure 11 polymers-14-02751-f011:**
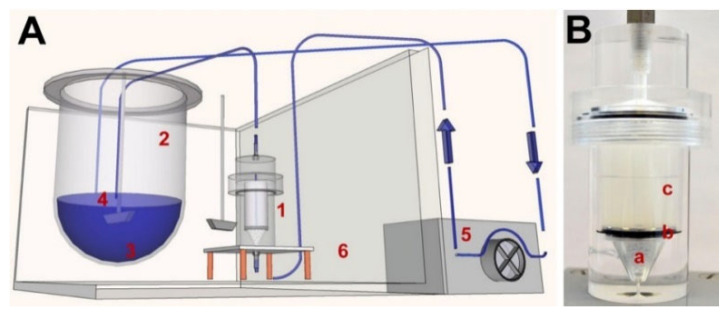
Schematic overview of the in vitro test setup (**A**) and photograph of the vFTC equipped with a 2 wt% agarose gel (**B**); (1) vFTC, (2) media container, (3) PBS of pH 7.4, (4) paddle stirred at 50 rpm, (5) peristaltic pump, (6) heated water bath, (a) glass beads, (b) stainless steel disc, (c) hydrogel, reproduced with permission from [59].

**Figure 12 polymers-14-02751-f012:**
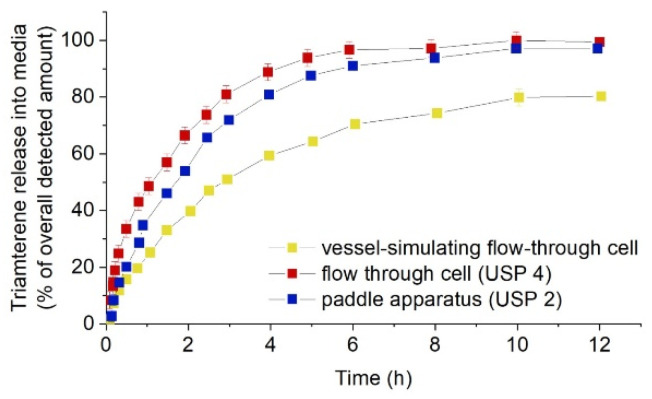
Normalized release of triamterene from stent coatings into media over time with vessel-simulating flow-through cell, flow-through cell (USP 4), or paddle apparatus (USP 2); flow rate 35 mL/min, paddle speed 50 rpm, reprinted from [85].

**Figure 13 polymers-14-02751-f013:**
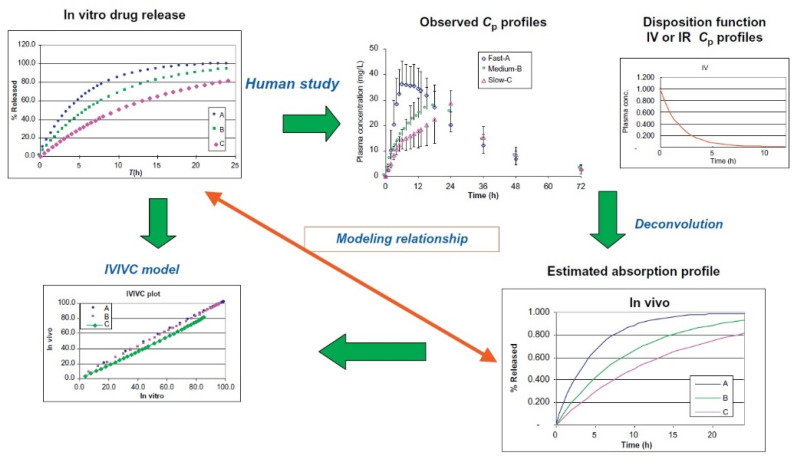
Illustration of convolution and deconvolution in IVIVC development, Reproduced with permission from [100]. Reproduced with permission from Y. Qiu, J.Z. Duan, Developing Solid Oral Dosage Forms; published by Elsevier, 2017.

**Figure 14 polymers-14-02751-f014:**
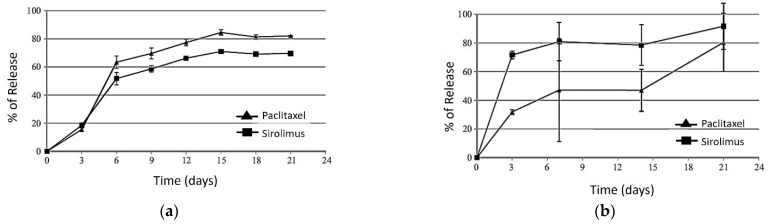
(**a**) In vitro and (**b**) in vivo, release profiles of paclitaxel and sirolimus from PLGA/ACP coated stents, reprinted from [81].

**Figure 15 polymers-14-02751-f015:**
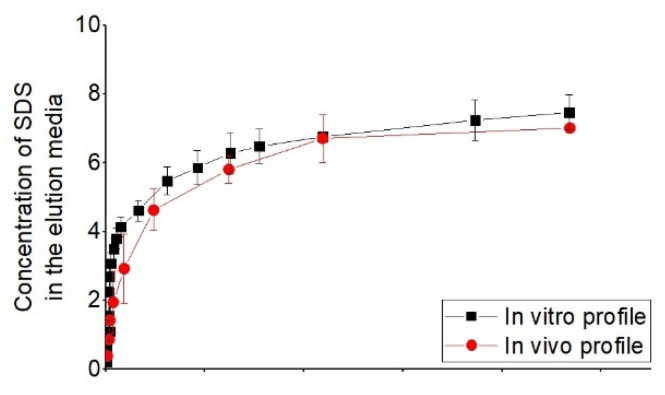
Comparison of the in vitro and in vivo profiles with time-scaling factor, reprinted from [56].

**Figure 16 polymers-14-02751-f016:**
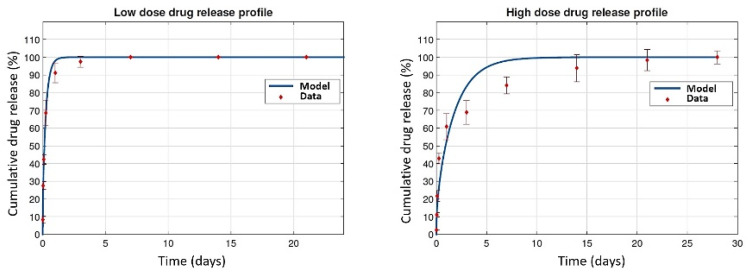
Comparison between experimental in vitro drug release data (normalized by the cumulative mass eluted by the final measurement time point), and model simulations. LEFT: The low dose stent was well-fitted by a diffusion coefficient of the order of 10^−16^ m^2^s^−1^. RIGHT: The fit between the diffusion model and the data from the high dose stent was not so good. Furthermore, the best-fitting diffusion coefficient was of the order 10^−17^ m^2^s^−1^, reprinted from [108]. Reproduced with permission from Craig M. McKittrick, Sean McKee, Simon Kennedy, Keith Oldroyd, Marcus Wheel, Giuseppe Pontrelli, Simon Dixon, Sean McGinty, Christopher McCormick, An Overview of In Vitro Drug Release Methods for Drug-Eluting Stents; published by Elsevier, 2019.

**Table 1 polymers-14-02751-t001:** Comparison of the artificial blood and tissue used in different in vitro studies.

Artificial Blood	Advantages/Dis-Advantages	Advantages/Dis-Advantages	Advantages/Dis-Advantages
phosphate buffered saline (PBS) [46,47]	pH constant 7.4, the proximity of its ions to the ions of the body (%)	calcium alginate hydrogel [58]	stability at 37 °C, the feasibility to adapt gel strength and elasticity, mild gelling conditions, feasibility of incorporation the diverse substances such as proteins or living cells
9:1 (*v*/*v*) of normal saline and isopropanol [53]	suitable medium for in vitro release of sirolimus	3 wt.% alginate; 2 wt.% agar; 2 wt.% agarose; 10 wt.% PAA; 15 wt.% PVA [59]	agarose: long-term dissolution
deionized water, PBS, phosphate-buffer (PB) [34]	deionized water increased the release compared to PBS and PB	calcium alginate; polyacrylamide (PAAm); poly(vinylethy limidazolium bromide [61]	disadvantages of calcium alginate: dissolution of the network by monovalent cations (like Na^+^) and its susceptibility to microbial contamination
surfactant 0.1% P123 (kind of PEO–PPO–PEO block copolymers) in phosphate buffer pH 4.0 [54]	suitable for in vitro release of sirolimus	alginate-based gel containing microparticles LiChroprep^®^ RP-18 or mediumchain triglycerides [64]	additives improved the transfer of hydrophobic drugs into the hydrogel but had no significant effect on the hydrophilic drugs
glycerol-water (40/60 vol%, 0.01% surfactant) [41,43]		deionized water, PBS, and PB as the base for preparing hydrogel [34]	more drug transfer to deionized waterbased hydrogels than PBS and PB-based hydrogels
2% ultra-pure sodium dodecyl sulfate (SDS), in high purity water with 10% gradient-grade acetonitrile (ACN), and buffered to pH 4.5 with phosphate+ 55:45:0.02 water/tetrahydrofuran (THF)/formic acid (*v*/*v*) [56]	good correlate between in vitro release profile with in vivo from porcine		
87% of glycerol and 13% of water [45]	approaching to the viscosity of blood		

## Data Availability

The data presented in this study are available on request from the corresponding author.

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
