# Peer review of "An Overview of In Vitro Drug Release Methods for Drug-Eluting Stents"

_polymers, 2022, doi:10.3390/polym14132751_

Round 1

Reviewer 1 Report

This review paper is especially important for several areas of knowledge; it brings relevant information and will contribute to the technical and scientific advances in pharmaceutical and medical sciences and material science.
I suggest revising and modifying the title; it does not seem appropriate. The methods reviewed for elution of drugs from stents were not bio-relevant.
Current methods have failed to ensure good IVIV correlation. This approach is very well supported in the manuscript. The authors were happy to show the drawbacks and the main challenges to overcome them, and both were appropriately discussed.
The illustrations were excellent, contributing to better compression of the text.
All abbreviations must be given in full the first time they appear in the text.
The conclusion seems a bit too pragmatic. The authors' perspective for a method that effectively measures the release profile and is bio-relevant for IVIV correlation is expected at the end of the findings.

Reviewer 2 Report

The manuscript by Abbasnezhad et al., entitled An overview on the Bio-relevant Methods of Drug Release Testing From Drug-Eluting Stents”, which aims to provide a comprehensive review of the methods used for in-vitro release testing of drug-eluting stents . This narrative review presents some aspects that need to be considered:

- I need to clarify the limitations of the review.

- The information that was discussed is of great importance, but it could be placed in tables and so the reader would have more clear information.

- In some figures the font size could be increased.

- It would be interesting to put the review highlights in the conclusion
